# MapTR: Structured Modeling and Learning for Online Vectorized HD Map Construction

**Bencheng Liao** [1,2,3] *   **Shaoyu Chen** [2,3] *   **Xinggang Wang** [2] †
**Tianheng Cheng** [2,3]   **Qian Zhang** [3]   **Wenyu Liu** [2]   **Chang Huang** [3]

[1] Institute of Artificial Intelligence, Huazhong University of Science & Technology
[2] School of EIC, Huazhong University of Science & Technology
[3] Horizon Robotics
`{bcliao,shaoyuchen,xgwang,thch,liuwy}@hust.edu.cn`
`{qian01.zhang, chang.huang}@horizon.ai`

## Abstract

High-definition (HD) map provides abundant and precise environmental information of the driving scene, serving as a fundamental and indispensable component for planning in autonomous driving system. We present MapTR, a structured end-to-end Transformer for efficient online vectorized HD map construction. We propose a unified permutation-equivalent modeling approach, *i.e.*, modeling map element as a point set with a group of equivalent permutations, which accurately describes the shape of map element and stabilizes the learning process. We design a hierarchical query embedding scheme to flexibly encode structured map information and perform hierarchical bipartite matching for map element learning. MapTR achieves the best performance and efficiency with only camera input among existing vectorized map construction approaches on nuScenes dataset. In particular, MapTR-nano runs at real-time inference speed (25.1 FPS) on RTX 3090, $8\times$ faster than the existing state-of-the-art camera-based method while achieving 5.0 higher mAP. Even compared with the existing state-of-the-art multi-modality method, MapTR-nano achieves 0.7 higher mAP , and MapTR-tiny achieves 13.5 higher mAP and $3\times$ faster inference speed. Abundant qualitative results show that MapTR maintains stable and robust map construction quality in complex and various driving scenes. MapTR is of great application value in autonomous driving. Code and more demos are available at `https://github.com/hustvl/MapTR`.

## 1 Introduction

High-definition (HD) map is the high-precision map specifically designed for autonomous driving, composed of instance-level vectorized representation of map elements (pedestrian crossing, lane divider, road boundaries, *etc.*). HD map contains rich semantic information of road topology and traffic rules, which is vital for the navigation of self-driving vehicle.

Conventionally HD map is constructed offline with SLAM-based methods (Zhang & Singh, 2014; Shan & Englot, 2018; Shan et al., 2020), incurring complicated pipeline and high maintaining cost. Recently, online HD map construction has attracted ever-increasing interests, which constructs map around ego-vehicle at runtime with vehicle-mounted sensors, getting rid of offline human efforts.

Early works (Chen et al., 2022a; Liu et al., 2021a; Can et al., 2021) leverage line-shape priors to perceive open-shape lanes based on the front-view image. They are restricted to single-view perception and can not cope with other map elements with arbitrary shapes. With the development of bird's eye view (BEV) representation learning, recent works (Chen et al., 2022b; Zhou & Krähenbühl, 2022; Hu et al., 2021; Li et al., 2022c) predict rasterized map by performing BEV semantic segmentation. However, the rasterized map lacks vectorized instance-level information, such as the lane structure,

---

* Equal contribution. † Corresponding author.

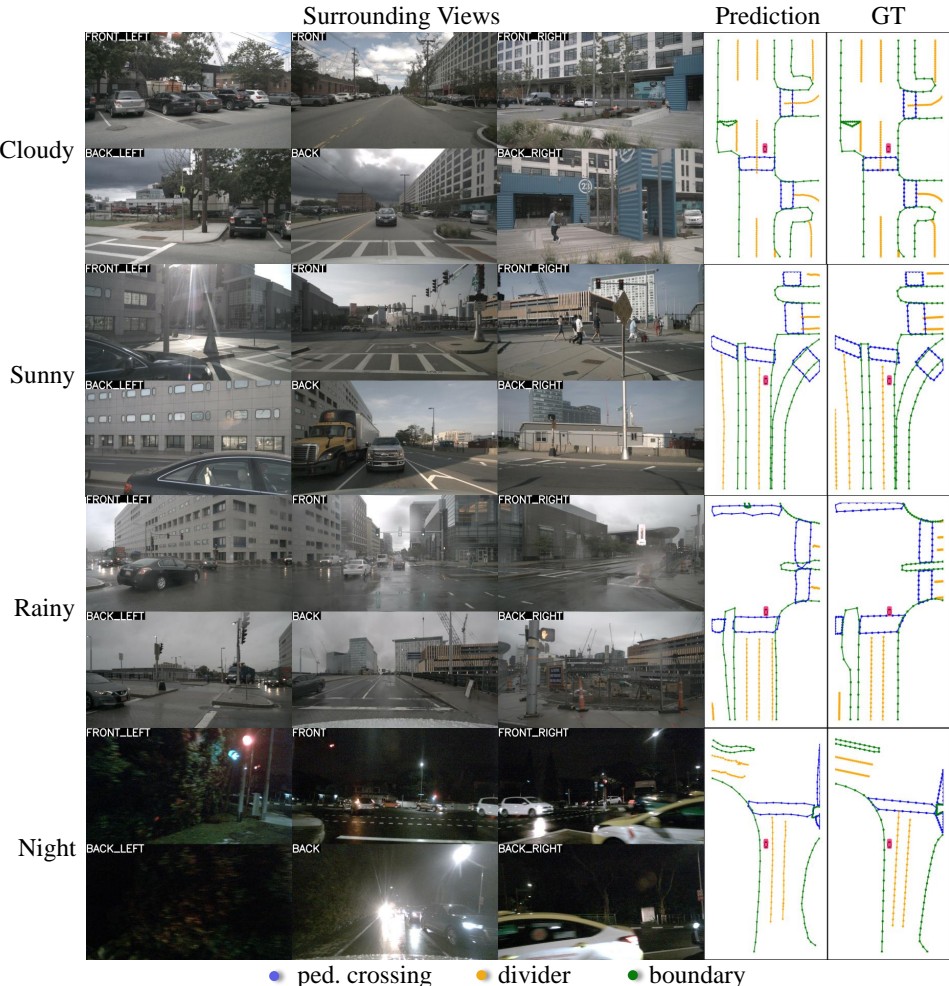

Figure 1. MapTR maintains stable and robust vectorized HD map construction quality in complex and various driving scenes.

which is important for the downstream tasks (*e.g.*, motion prediction and planning). To construct vectorized HD map, HDMapNet (Li et al., 2022a) groups pixel-wise segmentation results, which requires complicated and time-consuming post-processing. VectorMapNet (Liu et al., 2022a) represents each map element as a point sequence. It adopts a cascaded coarse-to-fine framework and utilizes auto-regressive decoder to predict points sequentially, leading to long inference time.

Current online vectorized HD map construction methods are restricted by the efficiency and not applicable in real-time scenarios. Recently, DETR (Carion et al., 2020) employs a simple and efficient encoder-decoder Transformer architecture and realizes end-to-end object detection.

It is natural to ask a question: *Can we design a DETR-like paradigm for efficient end-to-end vectorized HD map construction?* We show that the answer is affirmative with our proposed **Map TR**ansformer (MapTR).

Different from object detection in which objects can be easily geometrically abstracted as bounding box, vectorized map elements have more dynamic shapes. To accurately describe map elements, we propose a novel unified modeling method. We model each map element as a point set with a group of equivalent permutations. The point set determines the position of the map element. And the permutation group includes all the possible organization sequences of the point set corresponding to the same geometrical shape, avoiding the ambiguity of shape.

Based on the permutation-equivalent modeling, we design a structured framework which takes as input images of vehicle-mounted cameras and outputs vectorized HD map. We streamline the online vectorized HD map construction as a parallel regression problem. Hierarchical query embed-

dings are proposed to flexibly encode instance-level and point-level information. All instances and all points of instance are simultaneously predicted with a unified Transformer structure. And the training pipeline is formulated as a hierarchical set prediction task, where we perform hierarchical bipartite matching to assign instances and points in turn. And we supervise the geometrical shape in both point and edge levels with the proposed point2point loss and edge direction loss.

With all the proposed designs, we present MapTR, an efficient end-to-end online vectorized HD map construction method with unified modeling and architecture. MapTR achieves the best performance and efficiency among existing vectorized map construction approaches on nuScenes (Caesar et al., 2020) dataset. In particular, MapTR-nano runs at real-time inference speed (25.1 FPS) on RTX 3090, $8\times$ faster than the existing state-of-the-art camera-based method while achieving 5.0 higher mAP. Even compared with the existing state-of-the-art multi-modality method, MapTR-nano achieves 0.7 higher mAP and $8\times$ faster inference speed, and MapTR-tiny achieves 13.5 higher mAP and $3\times$ faster inference speed. As the visualization shows (Fig. 1), MapTR maintains stable and robust map construction quality in complex and various driving scenes.

Our contributions can be summarized as follows:

- We propose a unified permutation-equivalent modeling approach for map elements, *i.e.*, modeling map element as a point set with a group of equivalent permutations, which accurately describes the shape of map element and stabilizes the learning process.
- Based on the novel modeling, we present MapTR, a structured end-to-end framework for efficient online vectorized HD map construction. We design a hierarchical query embedding scheme to flexibly encode instance-level and point-level information, perform hierarchical bipartite matching for map element learning, and supervise the geometrical shape in both point and edge levels with the proposed point2point loss and edge direction loss.
- MapTR is the first real-time and SOTA vectorized HD map construction approach with stable and robust performance in complex and various driving scenes.

## 2 RELATED WORK

**HD Map Construction.** Recently, with the development of 2D-to-BEV methods (Ma et al., 2022), HD map construction is formulated as a segmentation problem based on surround-view image data captured by vehicle-mounted cameras. Chen et al. (2022b); Zhou & Krähenbühl (2022); Hu et al. (2021); Li et al. (2022c); Philion & Fidler (2020); Liu et al. (2022b) generate rasterized map by performing BEV semantic segmentation. To build vectorized HD map, HDMapNet (Li et al., 2022a) groups pixel-wise semantic segmentation results with heuristic and time-consuming post-processing to generate instances. VectorMapNet (Liu et al., 2022a) serves as the first end-to-end framework, which adopts a two-stage coarse-to-fine framework and utilizes auto-regressive decoder to predict points sequentially, leading to long inference time and the ambiguity about permutation. Different from VectorMapNet, MapTR introduces novel and unified modeling for map element, solving the ambiguity and stabilizing the learning process. And MapTR builds a structured and parallel one-stage framework with much higher efficiency.

**Lane Detection.** Lane detection can be viewed as a sub task of HD map construction, which focuses on detecting lane elements in the road scenes. Since most datasets of lane detection only provide single view annotations and focus on open-shape elements, related methods are restricted to single view. LaneATT (Tabelini et al., 2021) utilizes anchor-based deep lane detection model to achieve good trade-off between accuracy and efficiency. LSTR (Liu et al., 2021a) adopts the Transformer architecture to directly output parameters of a lane shape model. GANet (Wang et al., 2022) formulates lane detection as a keypoint estimation and association problem and takes a bottom-up design. Feng et al. (2022) proposes parametric Bezier curve-based method for lane detection. Instead of detecting lane in the 2D image coordinate, Garnett et al. (2019) proposes 3D-LaneNet which performs 3D lane detection in BEV. STSU (Can et al., 2021) represents lanes as a directed graph in BEV coordinates and adopts curve-based Bezier method to predict lanes from monocular camera image. Persformer (Chen et al., 2022a) provides better BEV feature representation and optimizes anchor design to unify 2D and 3D lane detection simultaneously. Instead of only detecting lanes in the limited single view, MapTR can perceive various kinds of map elements of $360°$ horizontal FOV, with a unified modeling and learning framework.

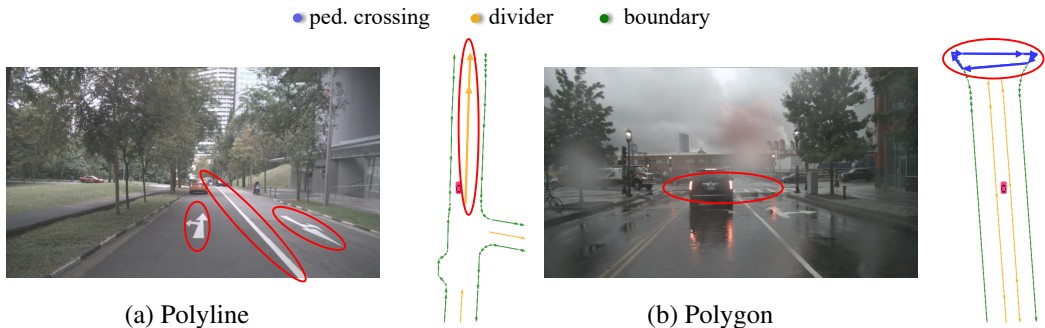

(a) Polyline          (b) Polygon

Figure 2. Typical cases for illustrating the ambiguity of map element about start point and direction. (a) Polyline: for the lane divider between two opposite lanes, defining its direction is difficult. Both endpoints of the lane divider can be regarded as the start point and the point set can be organized in two directions. (b) Polygon: for the pedestrian crossing, each point of the polygon can be regarded as the start point, and the polygon can be connected in two opposite directions (counter-clockwise and clockwise).

**Contour-based Instance Segmentation.** Another line of work related to MapTR is contour-based 2D instance segmentation (Zhu et al., 2022; Xie et al., 2020; Xu et al., 2019; Liu et al., 2021c). These methods reformulate 2D instance segmentation as object contour prediction task, and estimate the image coordinates of the contour vertices. CurveGCN (Ling et al., 2019) utilizes Graph Convolution Networks to predict polygonal boundaries. Lazarow et al. (2022); Liang et al. (2020); Li et al. (2021); Peng et al. (2020) rely on intermediate representations and adopt a two-stage paradigm, *i.e.*, the first stage performs segmentation / detection to generate vertices and the second stage converts vertices to polygons. These works model contours of 2D instance masks as polygons. Their modeling methods cannot cope with line-shape map elements and are not applicable for map construction. Differently, MapTR is tailored for HD map construction and models various kinds of map elements in a unified manner. Besides, MapTR does not rely on intermediate representations and has an efficient and compact pipeline.

## 3 MAPTR

### 3.1 PERMUTATION-EQUIVALENT MODELING

MapTR aims at modeling and learning the HD map in a unified manner. HD map is a collection of vectorized static map elements, including pedestrian crossing, lane divider, road boundarie, *etc.* For structured modeling, MapTR geometrically abstracts map elements as closed shape (like pedestrian crossing) and open shape (like lane divider). Through sampling points sequentially along the shape boundary, closed-shape element is discretized into polygon while open-shape element is discretized into polyline.

Preliminarily, both polygon and polyline can be represented as an ordered point set $V^F = [v_0, v_1, \ldots, v_{N_v-1}]$ (see Fig. 3 (Vanilla)). $N_v$ denotes the number of points. However, the permutation of the point set is not explicitly defined and not unique. There exist many equivalent permutations for polygon and polyline. For example, as illustrated in Fig. 2 (a), for the lane divider (polyline) between two opposite lanes, defining its direction is difficult. Both endpoints of the lane divider can be regarded as the start point and the point set can be organized in two directions. In Fig. 2 (b), for the pedestrian crossing (polygon), the point set can be organized in two opposite directions (counter-clockwise and clockwise). And circularly changing the permutation of point set has no influence on the geometrical shape of the polygon. Imposing a fixed permutation to the point set as supervision is not rational. The imposed fixed permutation contradicts with other equivalent permutations, hampering the learning process.

To bridge this gap, MapTR models each map element with $\mathcal{V} = (V, \Gamma)$. $V = \{v_j\}_{j=0}^{N_v-1}$ denotes the point set of the map element ($N_v$ is the number of points). $\Gamma = \{\gamma^k\}$ denotes a group of equivalent permutations of the point set $V$, covering all the possible organization sequences.

Specifically, for polyline element (see Fig. 3 (left)), $\Gamma$ includes 2 kinds of equivalent permutations:

$$\Gamma_{\text{polyline}} = \{\gamma^0, \gamma^1\} \begin{cases} \gamma^0(j) = j \mod N_v, \\ \gamma^1(j) = (N_v - 1) - j \mod N_v. \end{cases} \tag{1}$$

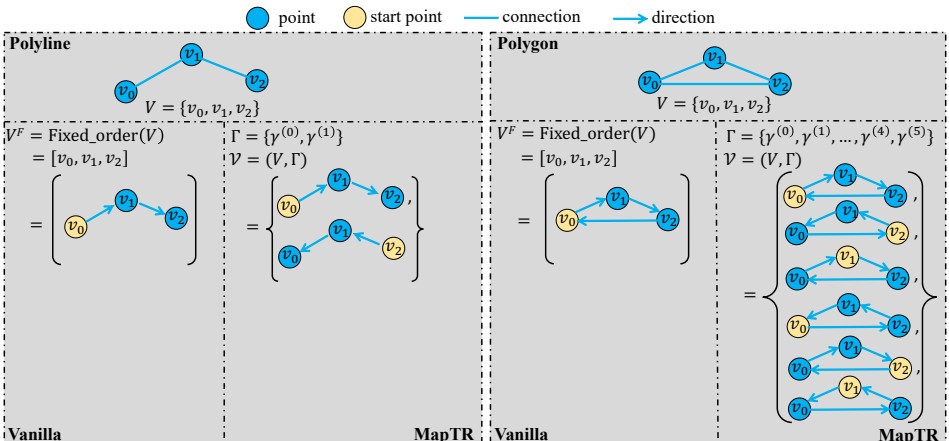

Figure 3. Illustration of permutation-equivalent modeling of MapTR. Map elements are geometrically abstracted and discretized into polylines and polygons. MapTR models each map element with $(V, \Gamma)$ (a point set $V$ and a group of equivalent permutations $\Gamma$), avoiding the ambiguity and stabilizing the learning process.

For polygon element (see Fig. 3 (right)), $\Gamma$ includes $2 \times N_v$ kinds of equivalent permutations:

$$
\Gamma_{\text{polygon}} = \{\gamma^0, \ldots, \gamma^{2 \times N_v - 1}\}
\begin{cases}
\gamma^0(j) = j \mod N_v, \\
\gamma^1(j) = (N_v - 1) - j \mod N_v, \\
\gamma^2(j) = (j + 1) \mod N_v, \\
\gamma^3(j) = (N_v - 1) - (j + 1) \mod N_v, \\
\ldots \\
\gamma^{2 \times N_v - 2}(j) = (j + N_v - 1) \mod N_v, \\
\gamma^{2 \times N_v - 1}(j) = (N_v - 1) - (j + N_v - 1) \mod N_v.
\end{cases}
\tag{2}
$$

By introducing the conception of equivalent permutations, MapTR models map elements in a unified manner and addresses the ambiguity issue. MapTR further introduces hierarchical bipartite matching (see Sec. 3.2 and Sec. 3.3) for map element learning, and designs a structured encoder-decoder Transformer architecture to efficiently predict map elements (see Sec. 3.4).

## 3.2 HIERARCHICAL MATCHING

MapTR parallelly infers a fixed-size set of $N$ map elements in a single pass, following the endto-end paradigm of query-based object detection and segmentation paradigm (Carion et al., 2020; Fang et al., 2021a;b). $N$ is set to be larger than the typical number of map elements in a scene. Let's denote the set of $N$ predicted map elements by $\hat{Y} = \{\hat{y}_i\}_{i=0}^{N-1}$. The set of ground-truth (GT) map elements is padded with $\varnothing$ (no object) to form a set with size $N$, denoted by $Y = \{y_i\}_{i=0}^{N-1}$. $y_i = (c_i, V_i, \Gamma_i)$, where $c_i$, $V_i$ and $\Gamma_i$ are respectively the target class label, point set and permutation group of GT map element $y_i$. $\hat{y}_i = (\hat{p}_i, \hat{V}_i)$, where $\hat{p}_i$ and $\hat{V}_i$ are respectively the predicted classification score and predicted point set. To achieve structured map element modeling and learning, MapTR introduces hierarchical bipartite matching, *i.e.*, performing instance-level matching and point-level matching in order.

**Instance-level Matching.** First, we need to find an optimal instance-level label assignment $\hat{\pi}$ between predicted map elements $\{\hat{y}_i\}$ and GT map elements $\{y_i\}$. $\hat{\pi}$ is a permutation of $N$ elements ($\hat{\pi} \in \Pi_N$) with the lowest instance-level matching cost:

$$
\hat{\pi} = \arg\min_{\pi \in \Pi_N} \sum_{i=0}^{N-1} \mathcal{L}_{\text{ins\_match}}(\hat{y}_{\pi(i)}, y_i).
\tag{3}
$$

$\mathcal{L}_{\text{ins\_match}}(\hat{y}_{\pi(i)}, y_i)$ is a pair-wise matching cost between prediction $\hat{y}_{\pi(i)}$ and GT $y_i$, which considers both the class label of map element and the position of point set:

$$
\mathcal{L}_{\text{ins\_match}}(\hat{y}_{\pi(i)}, y_i) = \mathcal{L}_{\text{Focal}}(\hat{p}_{\pi(i)}, c_i) + \mathcal{L}_{\text{position}}(\hat{V}_{\pi(i)}, V_i).
\tag{4}
$$

$\mathcal{L}_{\text{Focal}}(\hat{p}_{\pi(i)}, c_i)$ is the class matching cost term, defined as the Focal Loss (Lin et al., 2017) between predicted classification score $\hat{p}_{\pi(i)}$ and target class label $c_i$. $\mathcal{L}_{\text{position}}(\hat{V}_{\pi(i)}, V_i)$ is the position matching cost term, which reflects the position correlation between the predicted point set $\hat{V}_{\pi(i)}$ and the GT point set $V_i$ (refer to Sec. B for more details). Hungarian algorithm is utilized to find the optimal instance-level assignment $\hat{\pi}$ following DETR.

**Point-level Matching.** After instance-level matching, each predicted map element $\hat{y}_{\hat{\pi}(i)}$ is assigned with a GT map element $y_i$. Then for each predicted instance assigned with positive labels ($c_i \neq \varnothing$), we perform point-level matching to find an optimal point2point assignment $\hat{\gamma} \in \Gamma$ between predicted point set $\hat{V}_{\hat{\pi}(i)}$ and GT point set $V_i$. $\hat{\gamma}$ is selected among the predefined permutation group $\Gamma$ and with the lowest point-level matching cost:

$$\hat{\gamma} = \underset{\gamma \in \Gamma}{\arg\min} \sum_{j=0}^{N_v - 1} D_{\text{Manhattan}}(\hat{v}_j, v_{\gamma(j)}). \tag{5}$$

$D_{\text{Manhattan}}(\hat{v}_j, v_{\gamma(j)})$ is the Manhattan distance between the $j$-th point of the predicted point set $\hat{V}$ and the $\gamma(j)$-th point of the GT point set $V$.

## 3.3 TRAINING LOSS

MapTR is trained based on the optimal instance-level and point-level assignment ($\hat{\pi}$ and $\{\hat{\gamma}_i\}$). The loss function is composed of three parts, classification loss, point2point loss and edge direction loss:

$$\mathcal{L} = \lambda \mathcal{L}_{\text{cls}} + \alpha \mathcal{L}_{\text{p2p}} + \beta \mathcal{L}_{\text{dir}}, \tag{6}$$

where $\lambda$, $\alpha$ and $\beta$ are the weights for balancing different loss terms.

**Classification Loss.** With the instance-level optimal matching result $\hat{\pi}$, each predicted map element is assigned with a class label . The classification loss is a Focal Loss term formulated as:

$$\mathcal{L}_{\text{cls}} = \sum_{i=0}^{N-1} \mathcal{L}_{\text{Focal}}(\hat{p}_{\hat{\pi}(i)}, c_i). \tag{7}$$

**Point2point Loss.** Point2point loss supervises the position of each predicted point. For each GT instance with index $i$, according to the point-level optimal matching result $\hat{\gamma}_i$, each predicted point $\hat{v}_{\hat{\pi}(i),j}$ is assigned with a GT point $v_{i,\hat{\gamma}_i(j)}$. The point2point loss is defined as the Manhattan distance computed between each assigned point pair:

$$\mathcal{L}_{\text{p2p}} = \sum_{i=0}^{N-1} \mathbb{1}_{\{c_i \neq \varnothing\}} \sum_{j=0}^{N_v - 1} D_{\text{Manhattan}}(\hat{v}_{\hat{\pi}(i),j}, v_{i,\hat{\gamma}_i(j)}). \tag{8}$$

**Edge Direction Loss.** Point2point loss only supervises the node point of polyline and polygon, not considering the edge (the connecting line between adjacent points). For accurately representing map elements, the direction of the edge is important. Thus, we further design edge direction loss to supervise the geometrical shape in the higher edge level. Specifically, we consider the cosine similarity of the paired predicted edge $\hat{e}_{\hat{\pi}(i),j}$ and GT edge $e_{i,\hat{\gamma}_i(j)}$:

$$
\begin{aligned}
\mathcal{L}_{\text{dir}} &= -\sum_{i=0}^{N-1} \mathbb{1}_{\{c_i \neq \varnothing\}} \sum_{j=0}^{N_v - 1} \text{cosine\_similarity}(\hat{e}_{\hat{\pi}(i),j}, e_{i,\hat{\gamma}_i(j)}), \\
\hat{e}_{\hat{\pi}(i),j} &= \hat{v}_{\hat{\pi}(i),j} - \hat{v}_{\hat{\pi}(i),(j+1)\text{modN}_v}, \\
e_{i,\hat{\gamma}_i(j)} &= v_{i,\hat{\gamma}_i(j)} - v_{i,\hat{\gamma}_i(j+1)\text{modN}_v}.
\end{aligned}
\tag{9}
$$

## 3.4 ARCHITECTURE

MapTR designs an encoder-decoder paradigm. The overall architecture is depicted in Fig. 4.

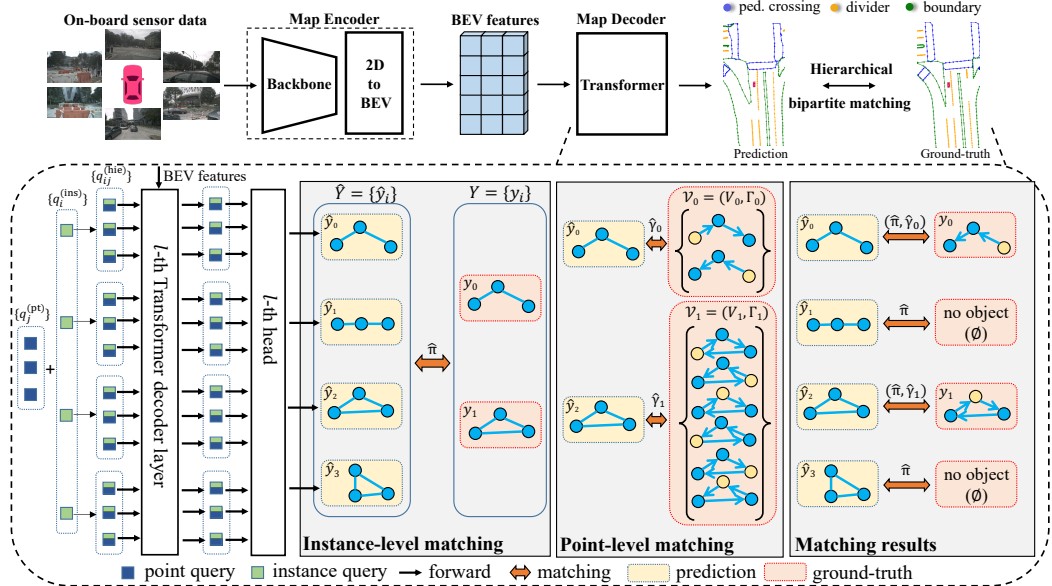

Figure 4. The overall architecture of MapTR. MapTR adopts an encoder-decoder paradigm. The map encoder transforms sensor input to a unified BEV representation. The map decoder adopts a hierarchical query embedding scheme to explicitly encode map elements and performs hierarchical matching based on the permutation-equivalent modeling. MapTR is fully end-to-end. The pipeline is highly structured, compact and efficient.

**Input Modality.** MapTR takes surround-view images of vehicle-mounted cameras as input. MapTR is also compatible with other vehicle-mounted sensors (*e.g.*, LiDAR and RADAR). Extending MapTR to multi-modality data is straightforward and trivial. And thanks to the rational permutation-equivalent modeling, even with only camera input, MapTR significantly outperforms other methods with multi-modality input.

**Map Encoder.** The map encoder of MapTR extracts features from images of multiple vehicle-mounted cameras and transforms the features into a unified feature representation, *i.e.*, BEV representation. Given multi-view images $\mathcal{I} = \{I_1, \ldots, I_K\}$, we leverage a conventional backbone to generate multi-view feature maps $\mathcal{F} = \{F_1, \ldots, F_K\}$. Then 2D image features $\mathcal{F}$ are transformed to BEV features $\mathcal{B} \in \mathbb{R}^{H \times W \times C}$. By default, we adopt GKT (Chen et al., 2022b) as the basic 2D-to-BEV transformation module, considering its easy-to-deploy property and high efficiency. MapTR is compatible with other transformation methods and maintains stable performance, *e.g.*, CVT (Zhou & Krähenbühl, 2022), LSS (Philion & Fidler, 2020; Liu et al., 2022c; Li et al., 2022b; Huang et al., 2021), Deformable Attention (Li et al., 2022c; Zhu et al., 2021) and IPM (Mallot et al., 1991). Ablation studies are presented in Tab. 4.

**Map Decoder.** We propose a hierarchical query embedding scheme to explicitly encode each map element. Specifically, we define a set of instance-level queries $\{q_i^{\text{ins}}\}_{i=0}^{N-1}$ and a set of point-level queries $\{q_j^{\text{pt}}\}_{j=0}^{N_v-1}$ shared by all instances. Each map element (with index $i$) corresponds to a set of hierarchical queries $\{q_{ij}^{\text{hie}}\}_{j=0}^{N_v-1}$. The hierarchical query of $j$-th point of $i$-th map element is formulated as:

$$q_{ij}^{\text{hie}} = q_i^{\text{ins}} + q_j^{\text{pt}}. \tag{10}$$

The map decoder contains several cascaded decoder layers which update the hierarchical queries iteratively. In each decoder layer, we adopt MHSA to make hierarchical queries exchange information with each other (both inter-instance and intra-instance). We then adopt Deformable Attention (Zhu et al., 2021) to make hierarchical queries interact with BEV features, inspired by BEVFormer (Li et al., 2022c). Each query $q_{ij}^{\text{hie}}$ predicts the 2-dimension normalized BEV coordinate $(x_{ij}, y_{ij})$ of the reference point $p_{ij}$. We then sample BEV features around the reference points and update queries.

Map elements are usually with irregular shapes and require long-range context. Each map element corresponds to a set of reference points $\{p_{ij}\}_{j=0}^{N_v-1}$ with flexible and dynamic distribution. The

reference points $\{p_{ij}\}_{j=0}^{N_v-1}$ can adapt to the arbitrary shape of map element and capture informative context for map element learning.

The prediction head of MapTR is simple, consisting of a classification branch and a point regression branch. The classification branch predicts instance class score. The point regression branch predicts the positions of the point sets $\hat{V}$. For each map element, it outputs a $2N_v$-dimension vector, which represents normalized BEV coordinates of the $N_v$ points.

## 4 EXPERIMENTS

**Dataset and Metric.** We evaluate MapTR on the popular nuScenes (Caesar et al., 2020) dataset, which contains 1000 scenes of roughly 20s duration each. Key samples are annotated at 2Hz. Each sample has RGB images from 6 cameras and covers $360°$ horizontal FOV of the ego-vehicle. Following the previous methods (Li et al., 2022a; Liu et al., 2022a), three kinds of map elements are chosen for fair evaluation – pedestrian crossing, lane divider, and road boundary. The perception ranges are $[-15.0m, 15.0m]$ for the $X$-axis and $[-30.0m, 30.0m]$ for the $Y$-axis. And we adopt average precision (AP) to evaluate the map construction quality. Chamfer distance $D_{Chamfer}$ is used to determine whether the prediction and GT are matched or not. We calculate the $\text{AP}_\tau$ under several $D_{Chamfer}$ thresholds ($\tau \in T, T = \{0.5, 1.0, 1.5\}$), and then average across all thresholds as the final AP metric:

$$\text{AP} = \frac{1}{|\text{T}|} \sum_{\tau \in \text{T}} \text{AP}_\tau. \tag{11}$$

**Implementation Details.** MapTR is trained with 8 NVIDIA GeForce RTX 3090 GPUs. We adopt AdamW (Loshchilov & Hutter, 2019) optimizer and cosine annealing schedule. For MapTR-tiny, we adopt ResNet50 (He et al., 2016) as the backbone. We train MapTR-tiny with a total batch size of 32 (containig 6 view images). All ablation studies are based on MapTR-tiny trained with 24 epochs. MapTR-nano is designed for real-time applications. We adopt ResNet18 as the backbone. More details are provided in Appendix A.

### 4.1 COMPARISONS WITH STATE-OF-THE-ART METHODS

In Tab. 1, we compare MapTR with state-of-the-art methods. MapTR-nano runs at real-time inference speed (25.1 FPS) on RTX 3090, $8\times$ faster than the existing state-of-the-art camera-based method (VectorMapNet-C) while achieving 5.0 higher mAP. Even compared with the existing state-of-the-art multi-modality method, MapTR-nano achieves 0.7 higher mAP and $8\times$ faster inference speed, and MapTR-tiny achieves 13.5 higher mAP and $3\times$ faster inference speed. MapTR is also a fast converging method, which demonstrate advanced performance with 24-epoch schedule.

| Method | Modality | Backbone | Epochs | $\text{AP}_{ped}$ | $\text{AP}_{divider}$ | $\text{AP}_{boundary}$ | mAP | FPS |
|---|---|---|---|---|---|---|---|---|
| HDMapNet | C | Effi-B0 | 30 | 14.4 | 21.7 | 33.0 | 23.0 | 0.8 |
| HDMapNet | L | PointPillars | 30 | 10.4 | 24.1 | 37.9 | 24.1 | 1.0 |
| HDMapNet | C & L | Effi-B0 & PointPillars | 30 | 16.3 | 29.6 | 46.7 | 31.0 | 0.5 |
| VectorMapNet | C | R50 | 110 | 36.1 | 47.3 | 39.3 | 40.9 | 2.9 |
| VectorMapNet | L | PointPillars | 110 | 25.7 | 37.6 | 38.6 | 34.0 | - |
| VectorMapNet | C & L | R50 & PointPillars | 110 | 37.6 | 50.5 | 47.5 | 45.2 | - |
| MapTR-nano | C | R18 | 110 | 39.6 | 49.9 | 48.2 | 45.9 | **25.1** |
| MapTR-tiny | C | R50 | **24** | 46.3 | 51.5 | 53.1 | 50.3 | 11.2 |
| MapTR-tiny | C | R50 | 110 | **56.2** | **59.8** | **60.1** | **58.7** | 11.2 |

Table 1. Comparisons with state-of-the-art methods (Liu et al., 2022a; Li et al., 2022a) on nuScenes `val` set. "C" and "L" respectively denotes camera and LiDAR. "Effi-B0" and "PointPillars" respectively correspond to Tan & Le (2019) and Lang et al. (2019). The APs of other methods are taken from the paper of VectorMapNet. The FPS of VectorMapNet-C is provided by its authors and measured on RTX 3090. Other FPSs are measured on the same machine with RTX 3090. "-" means that the corresponding results are not available. Even with only camera input, MapTR-tiny significantly outperforms multi-modality counterparts (+13.5 mAP). MapTR-nano achieves SOTA camera-based performance and runs at 25.1 FPS, realizing real-time vectorized map construction for the first time.

## 4.2  ABLATION STUDY

To validate the effectiveness of different designs, we conduct ablation experiments on nuScenes `val` set. More ablation studies are in Appendix B.

**Effectiveness of Permutation-equivalent Modeling.**  In Tab. 2, we provide ablation experiments to validate the effectiveness of the proposed permutation-equivalent modeling. Compared with vanilla modeling method which imposes a unique permutation to the point set, permutation-equivalent modeling solves the ambiguity of map element and brings an improvement of 5.9 mAP. For pedestrian crossing, the improvement even reaches 11.9 AP, proving the superiority in modeling polygon elements. We also visualize the learning process in Fig. 5 to show the stabilization of the proposed modeling.

| Modeling method | $AP_{ped}$ | $AP_{divider}$ | $AP_{boundary}$ | mAP |
|---|---|---|---|---|
| Fixed-order $V^F$ w/ ambiguity | 34.4 | 48.1 | 50.7 | 44.4 |
| Permutation-equivalent $(V, \Gamma)$ w/o ambiguity | 46.3 | 51.5 | 53.1 | 50.3 |

Table 2. Ablations about modeling method. Vanilla modeling method imposes a unique permutation to the point set, leading to ambiguity. MapTR introduces permutation-equivalent modeling to avoid the ambiguity, which stabilizes the learning process and significantly improves performance ( +5.9 mAP).

**Effectiveness of Edge Direction Loss.**  Ablations about the weight of edge direction loss are presented in Tab. 3. $\beta = 0$ means that we do not use edge direction loss. $\beta = 5e^{-3}$ corresponds to appropriate supervision and is adopted as the default setting.

| $\beta$ | $AP_{ped}$ | $AP_{divider}$ | $AP_{boundary}$ | mAP |
|---|---|---|---|---|
| 0 | 41.4 | 51.3 | 51.9 | 48.2 |
| $3e^{-3}$ | 44.8 | 50.4 | 52.1 | 49.1 |
| $5e^{-3}$ | 46.3 | 51.5 | 53.1 | 50.3 |
| $1e^{-2}$ | 41.9 | 50.9 | 52.0 | 48.3 |

Table 3. Ablations about the weight $\beta$ of edge direction loss.

| Method | mAP | FPS | Param. |
|---|---|---|---|
| IPM | 46.2 | 11.7 | 35.7M |
| LSS | 49.5 | 10.0 | 37.1M |
| Deform. Atten. | 49.7 | 11.2 | 36.0M |
| GKT | 50.3 | 11.2 | 35.9M |

Table 4. Ablations about 2D-to-BEV transformation methods. MapTR is compatible with various 2D-to-BEV methods and achieves stable performance.

**2D-to-BEV Transformation.**  In Tab. 4, we ablate on the 2D-to-BEV transformation methods (*e.g.*, IPM (Mallot et al., 1991), LSS (Liu et al., 2022c; Philion & Fidler, 2020), Deformable Attention (Li et al., 2022c) and GKT (Chen et al., 2022b)). We use an optimized implementation of LSS (Liu et al., 2022c). And for fair comparison with IPM and LSS, GKT and Deformable Attention both adopt one-layer configuration. Experiments show MapTR is compatible with various 2D-to-BEV methods and achieves stable performance. We adopt GKT as the default configuration of MapTR, considering its easy-to-deploy property and high efficiency.

## 4.3  QUALITATIVE VISUALIZATION

We show the predicted vectorized HD map results of complex and various driving scenes in Fig. 1. MapTR maintains stable and impressive results. More qualitative results are provided in Appendix C. We also provide videos (in the supplementary materials) to show the robustness.

## 5  CONCLUSION

MapTR is a structured end-to-end framework for efficient online vectorized HD map construction, which adopts a simple encoder-decoder Transformer architecture and hierarchical bipartite matching to perform map element learning based on the proposed permutation-equivalent modeling. Extensive experiments show that the proposed method can precisely perceive map elements of arbitrary shape in the challenging nuScenes dataset. We hope MapTR can serve as a basic module of self-driving system and boost the development of downstream tasks (*e.g.*, motion prediction and planning).

## ACKNOWLEDGMENT

This work was in part supported by NSFC (No. 6227072399). We would like to thank Yicheng Liu for his guidance on evaluation and constructive discussions.

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

# Appendix

## A    IMPLEMENTATION DETAILS

This section provides more implementation details of the method and experiments.

**Data Augmentation.**    The resolution of source images is $1600 \times 900$. For MapTR-nano, we resize the source images with $0.2$ ratio. For MapTR-tiny, we resize the source images with $0.5$ ratio. Color jitter is used by default.

**Model Setting.**    For all experiments, $\lambda$ is set to 2, $\alpha$ is set to 5, and $\beta$ is set to $5e^{-3}$ during training. For MapTR-tiny, we set the number of instance-level queries and point-level queries to 50 and 20 respectively. And we set the size of each BEV grid to $0.3m$ and stack 6 transformer decoder layers. We train MapTR-tiny with a total batch size of 32 (containig 6 view images), a learning rate of $6e^{-4}$, learning rate multiplier of the backbone is 0.1. All ablation studies are based on MapTR-tiny trained with 24 epochs. For MapTR-nano, we set the number of instance-level queries and point-level queries to 100 and 20 respectively. And we set the size of each BEV grid to $0.75m$ and stack 2 transformer decoder layers. We train MapTR-nano with 110 epochs, a total batch size of 192, a learning rate of $4e^{-3}$, learning rate multiplier of the backbone is 0.1. We employ GKT (Chen et al., 2022b) as the default 2D-to-BEV module of MapTR.

**Dataset Preprocessing.**    We process the map annotations following Liu et al. (2022a); Li et al. (2022a). Map elements in the perception ranges of ego-vehicle are extracted as ground-truth map elements. By default, The perception ranges are $[-15.0m, 15.0m]$ for the $X$-axis and $[-30.0m, 30.0m]$ for the $Y$-axis.

## B    ABLATION STUDY

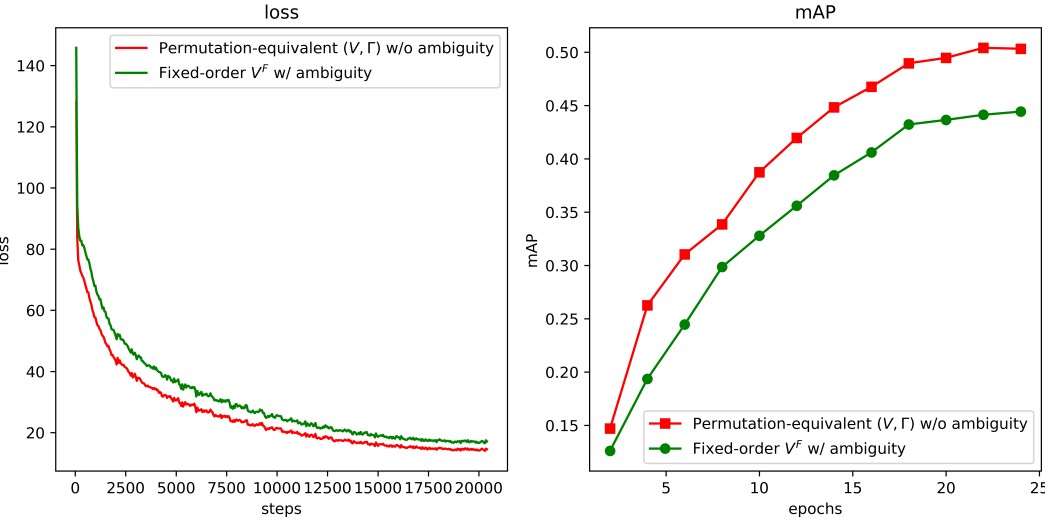

Figure 5. Convergence curves of permutation modeling methods .

**Point Number.**    Ablations about the number of points for modeling each map element are presented in Tab. 5. Too few points can not describe the complex geometrical shape of the map element. Too many points affect the efficiency. We adopt 20 points as the default setting of MapTR.

**Element Number.**    Ablations about the number of map elements are presented in Tab. 6. We adopt 50 as the default number of map elements for MapTR-tiny.

| Pt. num. | AP$_{ped}$ | AP$_{divider}$ | AP$_{boundary}$ | mAP | FPS |
|---|---|---|---|---|---|
| 10 | 42.5 | 51.3 | 50.1 | 48.0 | 12.3 |
| 20 | 46.3 | 51.5 | 53.1 | 50.3 | 11.2 |
| 40 | 44.7 | 52.4 | 52.9 | 50.0 | 10.8 |

Table 5. Ablations about the number of points for modeling each map element.

| Ele. num. | AP$_{ped}$ | AP$_{divider}$ | AP$_{boundary}$ | mAP | FPS |
|---|---|---|---|---|---|
| 25 | 36.3 | 43.8 | 44.7 | 41.6 | 11.4 |
| 50 | 46.3 | 51.5 | 53.1 | 50.3 | 11.2 |
| 75 | 48.2 | 53.1 | 55.3 | 52.2 | 11.1 |

Table 6. Ablations about the number of map elements.

**Decoder Layer Number.** Ablations about the layer number of map decoder are presented in Tab. 7. The map construction performance improves with more layers, but gets saturated when the layer number reaches 6.

| Layer num. | AP$_{ped}$ | AP$_{divider}$ | AP$_{boundary}$ | mAP | FPS |
|---|---|---|---|---|---|
| 1 | 20.8 | 30.2 | 36.3 | 29.1 | 15.2 |
| 2 | 36.0 | 43.1 | 48.0 | 42.4 | 14.2 |
| 3 | 38.2 | 44.1 | 49.5 | 44.0 | 13.5 |
| 6 | 46.3 | 51.5 | 53.1 | 50.3 | 11.2 |
| 8 | 39.6 | 51.9 | 51.2 | 47.6 | 10.6 |

Table 7. Ablations about the number of decoder.

**Position Matching Cost.** As mentioned in Sec. 3.2, we adopt the position matching cost term $\mathcal{L}_{\text{position}}(\hat{V}_{\pi(i)}, V_i)$ in instance-level matching, for reflecting the position correlation between the predicted point set $\hat{V}_{\pi(i)}$ and the GT point set $V_i$. In Tab. 8, we compare two kinds of cost design. *i.e.*, Chamfer distance cost and point2point cost. Point2point cost is similar to the point-level matching cost. Specifically, we find the best point2point assignment, and sum the Manhattan distance of all point pairs as the position matching cost of two point sets. The experiments show point2point cost is better than Chamfer distance cost.

| Position matching cost | AP$_{ped}$ | AP$_{divider}$ | AP$_{boundary}$ | mAP |
|---|---|---|---|---|
| Chamfer distance cost | 40.3 | 53.8 | 48.5 | 47.5 |
| Point2point cost | 46.3 | 51.5 | 53.1 | 50.3 |

Table 8. Ablations about the position matching cost term.

**Swin Transformer Backbones.** Ablations about the Swin Transformer backbones (Liu et al., 2021b) are presented in Tab. 9.

| Method | Backbone | AP$_{ped}$ | AP$_{divider}$ | AP$_{boundary}$ | mAP | FPS | Param. |
|---|---|---|---|---|---|---|---|
| MapTR-tiny | R50 | 46.3 | 51.5 | 53.1 | 50.3 | 11.2 | 35.9M |
| MapTR-tiny | Swin-tiny | 45.2 | 52.7 | 52.3 | 50.1 | 9.1 | 39.9M |
| MapTR-small | Swin-small | 50.2 | 55.4 | 57.3 | 54.3 | 7.3 | 61.2M |
| MapTR-base | Swin-base | 50.6 | 58.7 | 58.4 | 55.9 | 6.1 | 99.2M |

Table 9. Ablations about Swin Transformer backbones.

**Modality.** Multi-sensor perception is crucial for the safety of autonomous vehicles, and MapTR is compatible with other vehicle-mounted sensors like LiDAR. As illustrated in Tab. 10, with the schedule of only 24 epochs, multi-modality MapTR significantly outperform previous state-of-the-art result by 17.3 mAP while being 2× faster.

**Robustness to the camera deviation.** In real applications, the camera intrinsics are usually accurate and change little, but the camera extrinsics may be inaccurate due to the shift of camera position, calibration error, *etc*. To validate the robustness, we traverse the validation sets and randomly generate noise for each sample. We respectively add translation and rotation deviation of

| Method | Modality | Epochs | $AP_{ped}$ | $AP_{divider}$ | $AP_{boundary}$ | mAP | FPS |
|--------|----------|--------|-----------|----------------|-----------------|-----|-----|
| HDMapNet | C & L | 30 | 16.3 | 29.6 | 46.7 | 31.0 | 0.5 |
| VectorMapNet | C & L | 110 | 37.6 | 50.5 | 47.5 | 45.2 | <2.9 |
| MapTR-tiny | C | 24 | 46.3 | 51.5 | 53.1 | 50.3 | 11.2 |
| MapTR-tiny | L | 24 | 48.5 | 53.7 | 64.7 | 55.6 | 7.2 |
| MapTR-tiny | C & L | 24 | **55.9** | **62.3** | **69.3** | **62.5** | 5.8 |

Table 10. Ablations about the modality.

| Method | $\sigma_1(m)$ | | | | |
|--------|---|------|-----|-----|-----|
| | 0 | 0.05 | 0.1 | 0.5 | 1.0 |
| MapTR-tiny | 50.3 | 49.9 | 49.0 | 34.0 | 17.0 |

Table 11. Robustness to the translation deviation of camera. The metric is mAP. $\sigma_1$ is the standard deviation of $\Delta_x, \Delta_y, \Delta_z$.

| Method | $\sigma_2(rad)$ | | | | |
|--------|---|-------|------|------|------|
| | 0 | 0.005 | 0.01 | 0.02 | 0.05 |
| MapTR-tiny | 50.3 | 49.4 | 47.5 | 42.0 | 24.7 |

Table 12. Robustness to the rotation deviation of camera. The metric is mAP. $\sigma_2$ is the standard deviation of $\theta_x, \theta_y, \theta_z$.

different degrees. Note that we add noise to all cameras and all coordinates. And the noise is subject to normal distribution. There exists extremely large deviation in some samples, which affect the performance a lot. As illustrated in Tab. 11 and Tab. 12, when the standard deviation of $\Delta_x, \Delta_y, \Delta_z$ is $0.1m$ or the standard deviation of $\theta_x, \theta_y, \theta_z$ is $0.02rad$, MapTR still keeps comparable performance.

**Detailed running time.** To have a deeper understanding on the efficiency of MapTR, we present the detailed running time of each component in MapTR-tiny with only multi-camera input in Tab. 13.

| Component | Runtime (ms) | Proportion |
|-----------|--------------|------------|
| Backbone | 55.5 | 62.1% |
| 2D-to-BEV module (GKT) | 12.3 | 13.8% |
| Map decoder | 21.5 | 24.1% |
| Total | 89.3 | 100 % |

Table 13. Detailed running time for each component in MapTR-tiny on a RTX 3090.

## C  QUALITATIVE VISUALIZATION

We visualize map construction results of MapTR under various weather conditions and challenging road environment on nuScenes val set. As shown in Fig. 6, Fig. 7 and Fig. 8, MapTR maintains stable and impressive results. Video results are provided in the supplementary materials.

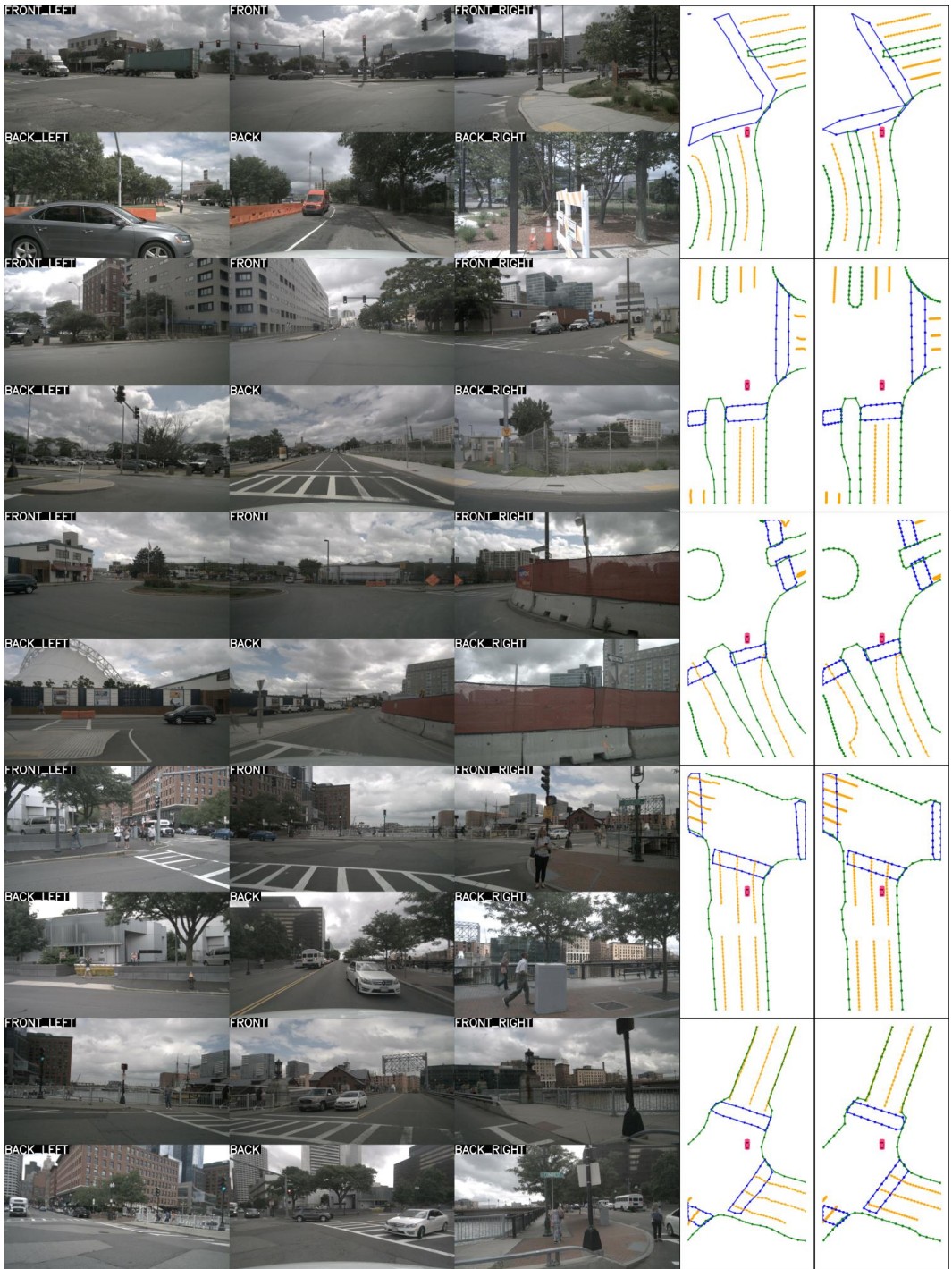

Figure 6. Visualization under sunny and cloudy weather.

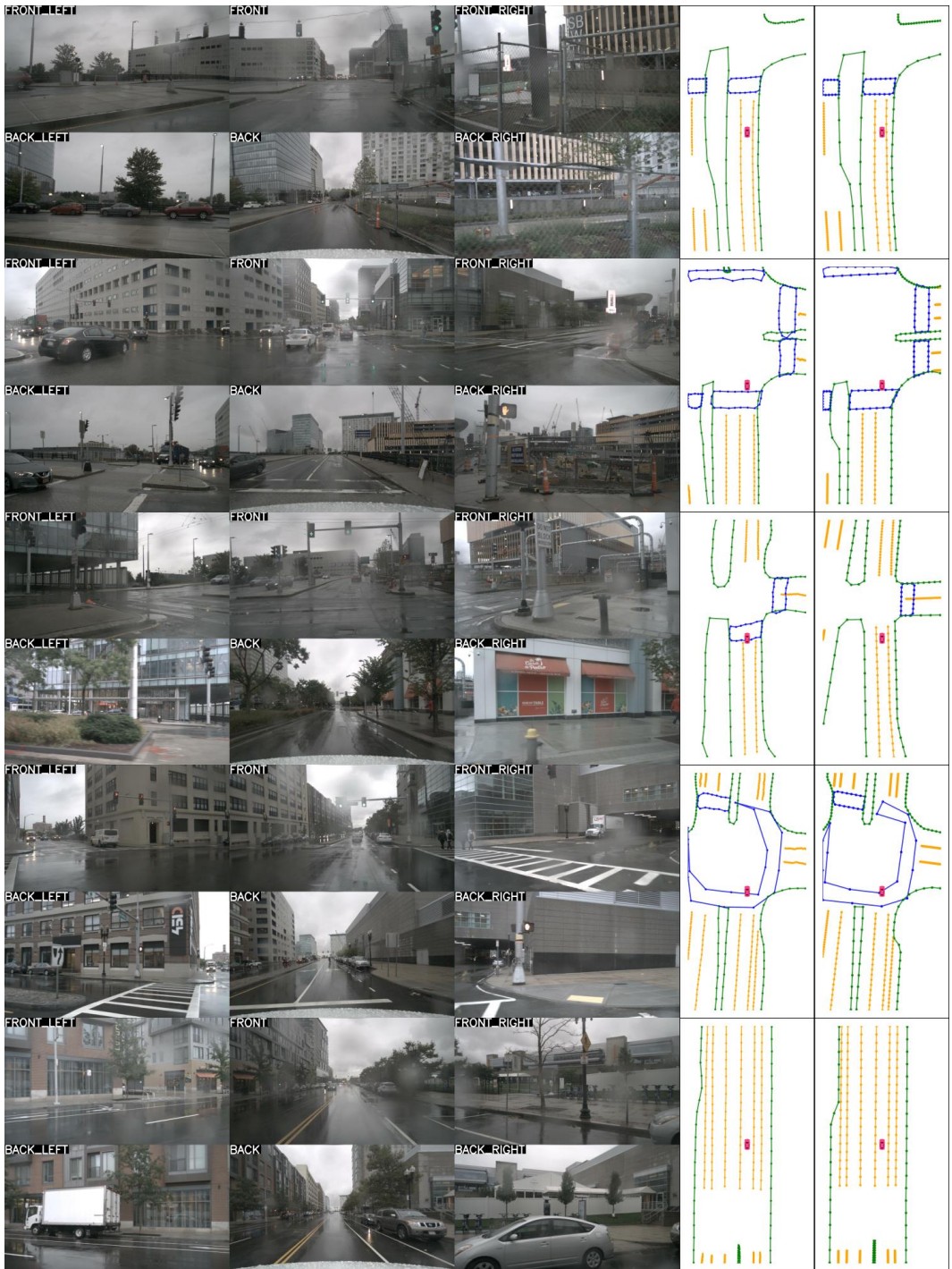

Figure 7. Visualization under rainy weather.

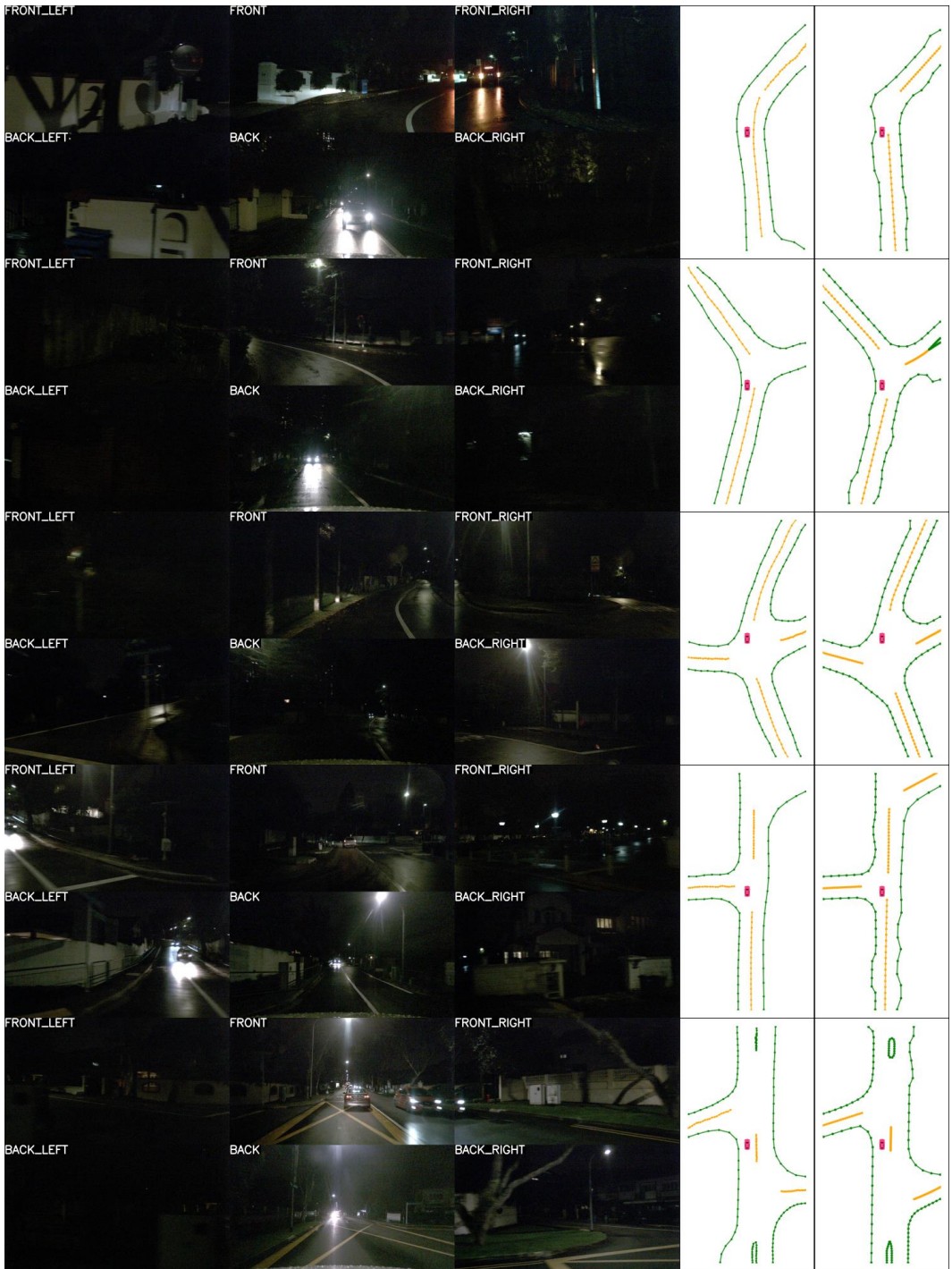

Figure 8. Visualization at night.

