# OpenReview forum: "MapTR: Structured Modeling and Learning for Online Vectorized HD Map Construction"
_ICLR.cc/2023/Conference — ICLR 2023 notable top 25%_

### Official Review · Reviewer_1muv · 2022-10-24

**Confidence:** 4
**Correctness:** 3
**Technical Novelty And Significance:** 2
**Empirical Novelty And Significance:** 3
**Recommendation:** 6

**Clarity, Quality, Novelty And Reproducibility:**

My main concerns are related to novelty and experiments. Clarity/paper issues are secondary.


1. Novelty: the permutation-invariant encoding alone is not enough for a new paper without proper vectorization experiments, see below.

2. Experiments: lacking information to determine the source of the performance gains; can you share some $\hat{V}$? why not add semantic segmentation + vectorization for comparison? It would show the superiority of your approach and require a minimal amount of changes for the decoder (segmentation already available in the GTK code). The question is, the ~mAP gain is gained from modeling? Is the information already available in the semantic segmentation and this is just a vectorization hat trick, or the vectorization method itself is the one improving performance?
Apart/in addition to this, adding other vectorization-only methods would help (see above).

3. Clarity/paper issues: again, a logic/consistency check is required. Some examples below:
- related work before conclusion? It is generally placed after the introduction and related to what follows, should be moved
- hollow claims - "stabilizes the learning process" - can you provide any graphs for that? Is it just an empirical observation?
- curious results -- Table 1, penultimate row, why add the results @epoch 24 and why continue the result at epoch 24?
- edge direction loss -- this one makes me think the modeling is not right; technically speaking we don't need it (since none of the polygons need ordering), it is an artifact of the proposed matching method; nevertheless, the frugal ablation study is related to weighting its loss weight ($\beta$) and turning on or off the permutation set
- text issues -- the E.g., "It is of great application
value in autonomous driving" -- adds nothing to the paper, repeated less than half a page away on page 3 and once again in the abstract, "equivalent permutations"/"permutation-equivalent" occurs 22 times [11+11], "8× faster" is repeated twice just in the abstract), small other issues - We adopt~s~ ResNet18, further research and application<s>
- figure 4 - "on board sensor data" >> explicitly state RGB cameras. Most sensors are generally on board, including LIDAR/IMUs. :)

Most of the information is available in order to reproduce the paper. The authors claim the code will be made public.

**Strength And Weaknesses:**

### Strengths:
- state-of-the-art results on nuScenes (~58.7 mAP, +8mAP)
- 8x more efficient processing compared to the previous contender (VectorMapNet)
- introduces modeling map element as a point set with a group of equivalent permutations

### Weaknesses:
- the vertex-to-polygon problem has a number of modelling options available in the literature (e.g., [BoundaryFormer, 1*, 2*-- the permutation issue doe]); apart from HDMapNet / VertexMapNet, none of them have been discussed; BoundaryFormer is summarily dismissed due to different domain.
- the speed claim can be largely attributed to GTK; given that GTK runs at 72.3 FPS on the 3090 (for the same dataset, probably no large setup differences), at least a proper timing table should be added (how much time is spent for encoder/decoder/matching)
- the paper could use a logic check, there are a number of redundant claims / language errors (see below)


[1*]Liang, J., Homayounfar, N., Ma, W. C., Xiong, Y., Hu, R., & Urtasun, R. (2020). Polytransform: Deep polygon transformer for instance segmentation. In Proceedings of the IEEE/CVF Conference on Computer Vision and Pattern Recognition (pp. 9131-9140).
[2*]Li, W., Zhao, W., Zhong, H., He, C., & Lin, D. (2021, May). Joint semantic-geometric learning for polygonal building segmentation. In Proceedings of the AAAI Conference on Artificial Intelligence (Vol. 35, No. 3, pp. 1958-1965).

**Summary Of The Paper:**

The paper describes a vectorization algorithm for map construction from surrounding views (RGB).


It acheives state-of-the art on the nuScenes dataset, surpasses multi-modality methods (RGB+LiDAR), for a final 58.7mAP (~ +8mAP over the previous method). A toned-down version of the algorithm (45.9mAP) reaches real time (~25FPS) on a high end GPU (RTX 3090).

The paper is essentially GKT (Chen et al., 2022b) + DETR (Carion et al., 2020) - style matching extended to point level + permutational encoding for vertex sequences. The speed gain is mostly due to GKT, the matching scheme is very similar to DETR, which leaves us with the main contribution: the permutational-equivalent encoding which adding all possible vertex sequences at training (~+6mAP gain).

**Summary Of The Review:**

Solid performance improvements for map vectorization on nuScenes (~+8%mAP), improved speed over predecessors, but under-investigated gains source (e.g., is it the vectorization method? is semantic segmentation still a better start, or we can drop it and just use vectorization, similar to BoundaryFormer, but keep the speed?) and frugal ablation study. This paper could have been very good; given the limited set of experiments, I cannot vouch for its future impact.

---

> ### Author Response · Authors · 2022-11-15
> **Thanks and Response to Reviewer 1muv (3/3)**
>
> $\color{red}{Question~5:}$ related work before conclusion? It is generally placed after the introduction and related to what follows, should be moved
>
> $\color{blue}{Response~5:}$ Thanks for the kind advice. In the revised paper, we have moved the section of related work to the proper place.
>
>
>  $\color{red}{Question~6:}$ hollow claims - "stabilizes the learning process" - can you provide any graphs for that? Is it just an empirical observation?
>
> $\color{blue}{Response~6:}$ It's not an empirical observation, but a well-founded conclusion.
> We provide an [anonymous link](https://user-images.githubusercontent.com/31960625/201886642-a019acd0-9fd2-4343-ba6c-8821e266a6db.png) to show the learning process. The curves show that, with the proposed modeling method which solves the ambiguity problem, the loss decreases much faster and the network better converges.
> Intuitively and theoretically speaking, if not adopting permutation-equivalent modeling, in the matching process of each optimization iteration, the network is randomly assigned with an arbitrary permutation of points. The target patterns for learning are inconsistent and the network is hard to converge. Hope these further explanations satisfy your concern.
>
>
> $\color{red}{Question~7:}$ curious results -- Table 1, penultimate row, why add the results @epoch 24 and why continue the result at epoch 24?
>
> $\color{blue}{Response~7:}$ We do not continue the result at epoch 24 to get result at epoch 110. Actually, @epoch 24 and @epoch 110 are two individual experiments, respectively corresponding to the short and long training schedules. We provide @epoch 24 results in order to show the superior convergence speed of MapTR.  MapTR trained for only 24 epochs achieves 9.4 higher mAP than VectorMapNet trained for 110 epochs (MapTR-tiny with only cameras, 50.3 mAP vs VectorMapNet with only cameras, 40.9 mAP ).
>
> $\color{red}{Question~8:}$ edge direction loss -- this one makes me think the modeling is not right; technically speaking we don't need it (since none of the polygons need ordering), it is an artifact of the proposed matching method; nevertheless, the frugal ablation study is related to weighting its loss weight (β) and turning on or off the permutation set.
>
> $\color{blue}{Response~8:}$ As  clarified above, MapTR is not a vertex-to-polygon algorithm. MapTR directly outputs an ordered point set for each map element. Edge direction loss is used to supervise the geometry of map element in the edge level. It makes sure the predicted edge has the consistent direction with the GT. Edge direction loss is reasonable but not an artifact. And we have presented thorough ablation studies. Some experiments are in Appendix, due to the page constraint.
>
>
> $\color{red}{Question~9:}$ text issues -- the E.g., "It is of great application value in autonomous driving" -- adds nothing to the paper, repeated less than half a page away on page 3 and once again in the abstract, "equivalent permutations"/"permutation-equivalent" occurs 22 times [11+11], "8× faster" is repeated twice just in the abstract), small other issues - We adopts ResNet18, further research and application\<s\>
>
> $\color{blue}{Response~9:}$ Thanks for the helpful advice. We have removed the duplicated statements and corrected the typos in the revised paper.
>
> $\color{red}{Question~10:}$ figure 4 - "on board sensor data" >> explicitly state RGB cameras. Most sensors are generally on board, including LIDAR/IMUs. :)
>
> $\color{blue}{Response~10:}$ We state "on board sensor data" under the consideration that MapTR is  also compatible with other vehicle-mounted sensors like LiDAR. In Table R6, we present the results of MapTR based on different sensor data.  Though trained for only 24 epochs, multi-modality results significantly outperform previous SOTA results by 17.3 mAP.  And higher performance is expected if trained for 110 epochs.
>
> | Method | Modality | Epochs| AP$_{ped}$|AP$_{divider}$|AP$_{boundary}$| mAP| FPS|
> | :---: | :---: |:---: |:---: |:---: |:---: |:---: |:---:|
> | MapTR-tiny   |Camera  | 24 |46.3|51.5|53.1|50.3|11.2
> | MapTR-tiny   |LiDAR  | 24 |48.5 |53.7|64.7|55.6|7.2
> | MapTR-tiny   |Camera&LiDAR  | 24 |55.9 |62.3|69.3|62.5|5.8|
>
> Table R7: Results of MapTR based on different onboard sensor data.
>
>
> $\color{red}{Question~11:}$ Most of the information is available in order to reproduce the paper. The authors claim the code will be made public.
>
> $\color{blue}{Response~11:}$ We provide the complete code with guidance for training and evaluation in the supplementary materials. And we promise that the results are totally reproducible and the source code can be publicly accessible.
>
> If the clarifications above satisfy your concerns, please consider reassessing this paper. Thanks for your valuable comments and we are expecting further discussions with you.

---

> ### Author Response · Authors · 2022-11-15
> **Thanks and Response to Reviewer 1muv (2/3)**
>
>
> $\color{red}{Question~3:}$ the speed claim can be largely attributed to GKT; given that GKT runs at 72.3 FPS on the 3090 (for the same dataset, probably no large setup differences), at least a proper timing table should be added (how much time is spent for encoder/decoder/matching)
>
> $\color{blue}{Response~3:}$ As clarified above, in MapTR,  GKT is only a Transformer layer for transforming 2D image features to BEV features. We do not use other parts of GKT. To response to your concerns, we provide the timing table in Table R5. Backbone accounts for most of the inference time (62.2%). MapTR takes high-resolution multi-view images (1600 × 900 x 6 views) as input and thus backbone is time-consuming. GKT (for 2D-to-BEV transformation) is a Transformer layer and only accounts for 13.8% of the total inference time. The efficiency of MapTR is mainly attributed to the efficient map decoder, which only takes 21.5 ms (24.1% of the inference time). Matching is only used in the training phase, and does not affect the inference speed.
>
>
> | Component        | Backbone | 2D-to-BEV (GKT) | Map Decoder |  Matching | Total |
> | ------ | ------ | ------ | ------ | ------ | ------ |
> |  Inference time   |  55.5 ms (62.1%) |   12.3 ms (13.8%)  | 21.5 ms (24.1%) |   0.0 ms (0.0%) |  89.3 ms (100%)|
>
>
> Table R5: The inference time of different components of MapTR-Tiny.
>
>
>
>
>
> $\color{red}{Question~4:}$ Novelty: the permutation-invariant encoding alone is not enough for a new paper without proper vectorization experiments, see below.
> Experiments: lacking information to determine the source of the performance gains; can you share some V^? why not add semantic segmentation + vectorization for comparison? It would show the superiority of your approach and require a minimal amount of changes for the decoder (segmentation already available in the GTK code). The question is, the ~mAP gain is gained from modeling? Is the information already available in the semantic segmentation and this is just a vectorization hat trick, or the vectorization method itself is the one improving performance? Apart/in addition to this, adding other vectorization-only methods would help (see above).
>
> $\color{blue}{Response~4:}$ As clarified above, the novelties of this paper are much more than permutation-equivalent modeling. The proposed hierarchical embedding, the manner in which hierarchical map queries interact and the overall architecture design are also our contributions.
> MapTR is not a incremental vectorization method based on GKT, but a brand-new framework for vectorized map construction. We directly use map queries to get features from BEV feature maps and output vectorized map instances. The source of the performance gains also comes from overall architecture design.
>
> To further demonstrate the difference between MapTR's framework and semantic segmentation + vectorization framework (e.g., HDMapNet), we take the results from the original paper to form Table R6 below. We replace GKT with IPM for 2D-to-BEV transformation to show that MapTR relies little on GKT. As illustrated in Table R6, the SOTA speed and results  come from the novel modeling and efficient one-stage designs.  And in the original paper, we provide thorough experiments to validate the effectiveness of framework design. Due to the limited space, we move some  experiments to the Appendix.
>
> MapTR doesn't need semantic segmentation and doesn't have intermediate results like segmented vertex or bounding box. Vectorization-only methods can not be applied on MapTR for comparison. Other methods you have mentioned ([1*], [2*] and BoundaryFormer) can't cope with line-shape map elements and is not applicable for map construction. Comparing MapTR with them is infeasible.
>
>
>
> | Method | Modality  | 2D-to-BEV|Epochs| Framework| mAP| FPS|
> | :---:  | :---:    |:---:     |:---: |  :---:   |:---: |:---: |
> | HDMapNet$^{*}$   |Camera  | IPM |30| semantic segmentaion + vectorization|23.0|0.8|
> | VectorMapNet$^{*}$   |Camera  | IPM |110|two-stage, end-to-end, autoregressive|45.2|2.9|
> | MapTR-tiny$^{\dagger}$   |Camera  | IPM |**24**| **one-stage, end-to-end,  permutation-equivalent**|**46.2**|**11.7**|
>
> Table R6: Comparsions of different modeling methods. $^{*}$ means the results are taken from Table 1 in the original paper, $^{\dagger}$ means the result is taken from Table 4. And higher performance is expected if MapTR-tiny with IPM is trained for 110 epochs.

---

> ### Author Response · Authors · 2022-11-15
> **Thanks and Response to Reviewer 1muv (1/3)**
>
> Thanks for your valuable comments and suggestions which help us improve this work. We'd like to first provide some clarifications about MapTR.
>
> $\color{blue}{Clarification:}$ MapTR is not a combination of GKT and DETR-like style matching.  It's not segmentation + vertex-to-polygon vectorization algorithm. In fact, MapTR is fully end-to-end: It takes RGB images as input and output vectorized maps, without any intermediate representations (e.g., segmentation results). In MapTR, GKT is only a Transformer layer for transforming 2D image features to BEV features. MapTR relies little on GKT.  We can even replace GKT with a simple Inverse Perspective Mapping (IPM) operation, and achieve similar speed and performance (please refer to Table 4 of the paper). Thus, the speed claim is not attributed to GKT at all, but attributed to the overall efficient design of MapTR (without any post-processing steps, only one-stage, without intermediate representations, etc.). Besides, though MapTR is inspired by DETR, the hierarchical matching scheme is novel and effective, which is specially designed for map construction.
>
> Following are our detailed responses to your concerns.
>
> $\color{red}{Question\ 1:}$  The paper is essentially GKT (Chen et al., 2022b) + DETR (Carion et al., 2020) - style matching extended to point level + permutational encoding for vertex sequences. The speed gain is mostly due to GKT, the matching scheme is very similar to DETR, which leaves us with the main contribution: the permutational-equivalent encoding which adding all possible vertex sequences at training (~+6mAP gain).
>
> $\color{blue}{Response~1:}$ As clarified above, MapTR is not GKT + DETR  and  the speed gain is not due to GKT. The contributions of this work are much more than permutational-equivalent modeling. We also propose the hierarchical embedding scheme to flexibly encode map elements and design how hierarchical map queries interact. Besides, MapTR is a novel end-to-end framework and the overall architecture design is also our contribution.
>
>
> $\color{red}{Question~2:}$ The vertex-to-polygon problem has a number of modelling options available in the literature (e.g., [BoundaryFormer, 1*, 2*-- the permutation issue doe]); apart from HDMapNet / VertexMapNet, none of them have been discussed; BoundaryFormer is summarily dismissed due to different domain.
>
> [1*]Liang, J., Homayounfar, N., Ma, W. C., Xiong, Y., Hu, R., & Urtasun, R. (2020). Polytransform: Deep polygon transformer for instance segmentation. In Proceedings of the IEEE/CVF Conference on Computer Vision and Pattern Recognition (pp. 9131-9140).
>
> [2*]Li, W., Zhao, W., Zhong, H., He, C., & Lin, D. (2021, May). Joint semantic-geometric learning for polygonal building segmentation. In Proceedings of the AAAI Conference on Artificial Intelligence (Vol. 35, No. 3, pp. 1958-1965).
>
>
> $\color{blue}{Response~2:}$ MapTR is quite different from [1*], [2*] and BoundaryFormer. [1*], [2*] and BoundaryFormer are based on intermediate results and are two-stage. The first stage performs segmentation/detection to generate vertices.  The second stage converts vertices to polygons. MapTR is end-to-end and one-stage, i.e., taking RGB images as input and directly outputting vectorized maps, without any intermediate representations. And the modelling methods used in [1*], [2*] and BoundaryFormer are not applicable for modelling map elements. Most map elements are line-shape (especially for lane divider and road curb). [1*], [2*] and BoundaryFormer are proposed to model the 2D instance masks and can't cope with line-shape map elements. We have added more discussions about these works in revision following your suggestion.

---

> ### Author Response · Authors · 2022-12-02
> **A gentle reminder for Reviewer 1muv**
>
> Dear Reviewer 1muv ,
>
> We appreciate your constructive comments for helping us improve our paper in many aspects. This is a gentle reminder since the final discussion stage ends soon.
>
> We would like to ask if Reviewer 1muv may have any further questions regarding our submission so that we can still respond. If you are satisfied with our response and changes, please consider updating the rating.
>
> Best wishes,
>
> Authors

---

> ### Comment · Reviewer_1muv · 2022-12-06
> **Thank you!**
>
> Dear authors,
>
>
> Thank you for the clarifications!
>
> R1,3,4 -- I stand corrected with the impact of GTK, I was wrong to assume it had a greater impact and it would be nice to include the numbers you have provided me in the final version of the paper.
>
> R2 -- I would argue that the end result is more relevant than the number of stages from the pipeline. A polygon could be collapsed into a line and run through a final pruning in order to remove redundant vertices. I would still find the comparison relevant and needed. Maybe you could combine an intermediate representation method with yours, sort of an attention mechanism. The polygon stability at small speeds / near static scenes is not that good and it could probably be improved with this approach.
>
> R6 - thank you, please add the Table in the supplementary.
>
> R7 - ok, it was not clear from the text.
>
> R8 - I understand that, I was suggesting that there is no 'standard' direction to follow a contour and that it is an artificial constraint derived from your problem modelling.
>
> R9 - Interesting, adding lidar worsens the results for crossing... It would be insightful to also add those results, IMHO.
>
> There is a combination of marketing/overclaims, curious design decisions and comparison stiffness (i.e., 2 stage methods are obsolete) that prevent me from upgrading the rating. Nevertheless, I still recommend accepting the paper, as I originally did.
>
>
> All the best,
> 1muv

---

> > ### Author Response · Authors · 2022-12-08
> > **Thanks for the feedback**
> >
> > We are glad to have addressed your concerns. Thanks for your feedback and we will improve our manuscript accordingly.
> >
> > >  I would argue that the end result is more relevant than the number of stages from the pipeline. A polygon could be collapsed into a line and run through a final pruning in order to remove redundant vertices. I would still find the comparison relevant and needed. Maybe you could combine an intermediate representation method with yours, sort of an attention mechanism. The polygon stability at small speeds / near static scenes is not that good and it could probably be improved with this approach.
> >
> > Thanks for your suggestion. In this paper,  we focus on an end-to-end and unified one-stage design. MapTR can also be extended to a two-stage design for better performance. As for how to utilize intermediate representation and post-processings to improve the quality, we agree it would be an interesting and promising direction to explore in the future. And we hope that MapTR lays a good foundation for future work.
> >
> > >  I was suggesting that there is no 'standard' direction to follow a contour and that it is an artificial constraint derived from your problem modeling.
> >
> > We agree with you that there is no  'standard' direction to follow a contour. However, we do need a direction to organize the point set into contour.
> > Directly using annotated direction is suboptimal. So MapTR proposes the permutation-equivalent modeling, which automatically selects an optimal direction. And the edge-direction loss is built upon the automatically selected direction.

---

### Official Review · Reviewer_oVMw · 2022-10-25

**Confidence:** 4
**Correctness:** 3
**Technical Novelty And Significance:** 3
**Empirical Novelty And Significance:** 4
**Recommendation:** 6

**Clarity, Quality, Novelty And Reproducibility:**

The quality of this paper is good -- it presents the key idea clearly, proposes several novel approaches to the issues of previous map learning frameworks, shows strong empirical performance. I do have some concerns about the reproducibility. So I encourage authors to add more details of the model and open-source code after the review process.

**Strength And Weaknesses:**

Strength:

1. This paper further streamlines recent learning-based map learning approaches and proposes a fully end-to-end solution to learning an online HD map from multi-camera images and point clouds. This pipeline is simple and effective.

2. The proposed hierarchical modeling of map elements is technically sound. This follows a coarse-to-fine paradigm and model map elements in both instance level and point level. Such operation not only improves stability and performance of map learning as studied in this paper, but also has potentials to benefit other tasks that also use a Transformer-based architectures.

3. This paper also introduced a permutation equivalent module to model map elements, which I think is novel. This permutation equivalent module removes the ambiguity introduced by primitives like polylines and polygons.

4. This paper demonstrates strong empirical performance on several benchmarks, compared to existing methods. Also, this paper also shows that the proposed method can achieve real-time performance and be developed to cars.

Weaknesses:

1. My first concern is the reproducibility. The proposed method seems to be a complex Transformer model with several new modules. With current level of details, I am not sure if one can faithfully reproduce this work and achieve similar performance.

2. It remains an open question to choose between polylines and polygons if we consider new map elements. Can this paper present some guidances of that? Can we unify the polyline representations and the polygon representations for all map elements?

3. More ablation studies are needed. E.g., the choices of polylines and polygons as stated above.

4. This paper (arguably) overclaims several contributions. It is *not* the first end-to-end approach as far as I know (e.g. VectorMapNet already introduced the Transformer-based end-to-end pipeline).

**Summary Of The Paper:**

This paper introduces a novel pipeline for HD map learning, which is an essential component of autonomous driving. Traditional methods use pre-annotated HD maps for localization and mapping, preventing autonomous driving scaling up. To address that, recent works aim to predict the HD maps on-the-fly with machine learning models. These works, however, require some hand-crafted post-processing steps. This paper proposes a fully end-to-end approach to HD map learning, based on recent DETR architecture. In addition to the architecture, this paper also studies how to effectively model map elements and introduces a permutation equivalent modeling of map elements. Under all settings, this paper shows significant improvements over baselines (HDMapNet & VectorMapNet).

**Summary Of The Review:**

Overall, I like this paper and recommend weakly acceptance though with several concerns (I believe they can be addressed in the rebuttal). I think this paper proposes a simple and effective pipeline for this online map learning problem. It presents clear motivation and strong empirical performance. That said, I am looking forward to seeing authors' comments to my concerns.

---

> ### Author Response · Authors · 2022-11-15
> **Thanks and Response to Reviewer oVMw**
>
> Thanks for your valuable comments and suggestions. We’d like to first provide some clarifications about MapTR.
>
> $\color{blue}{Clarification:}$ MapTR is a unified modeling framework for map elements in arbitrary shapes. We distinguish polyline and polygon **only in training**. Because in hierarchical matching, polyline and polygon correspond to different equivalent permutations and should be processed differently. During inference, polygon and polyline are treated in the unified manner. Specifically, for each map element, MapTR directly outputs a point list whose permutation is automatically chosen by the model without knowing whether it is polygon or polyline in advance. After inference, according the connectivity, we can easily judge whether the predicted map element is polygon or polyline. I.e., if the first point and the last point of the predicted point list correspond to the same location, then the map element is closed-shape and thus is a polygon. Otherwise, the predicted map element is open-shape and thus is a polyline.
>
> Following are our detailed responses to your concerns.
>
>
> $\color{red}{Question~1:}$ My first concern is the reproducibility. The proposed method seems to be a complex Transformer model with several new modules. With current level of details, I am not sure if one can faithfully reproduce this work and achieve similar performance.
>
> $\color{blue}{Response~1:}$ To satisfy your concern about the reproducibility,  in the supplementary material, we provide the complete code with guidelines for training and evaluation. And we promise the reproducibility of MapTR.
>
> $\color{red}{Question~2:}$ It remains an open question to choose between polylines and polygons if we consider new map elements. Can this paper present some guidances of that? Can we unify the polyline representations and the polygon representations for all map elements?
> More ablation studies are needed. E.g., the choices of polylines and polygons as stated above.
>
> $\color{blue}{Response~2:}$ As stated in $\color{blue}{Clarification}$, during inference, we do unify the polyline representations and the polygon representations for all map elements. Polygon and polyline are treated in the unified manner. For each map element, MapTR directly outputs a point list whose permutation is automatically chosen by the model without knowing whether it is polygon or polyline in advance. And we provide the code in the supplementary material. All the implementation details about polygon and polyline are available.
>
>
> $\color{red}{Question~3:}$ This paper (arguably) overclaims several contributions. It is not the first end-to-end approach as far as I know (e.g. VectorMapNet already introduced the Transformer-based end-to-end pipeline).
>
> $\color{blue}{Response~3:}$ Thanks for your kind suggestion. We want to point it out that we did not and do not claim that MapTR is the first end-to-end approach. As stated in the original paper, we claim that MapTR is the first **real-time**  and SOTA vectorized HD map construction approach as proved in Table 1. VectorMapNet is not real-time. We will clarify our contribution more clearly in revision in case of misunderstanding.
>
> If the response satisfies your concerns, please consider reassessing the score. Thanks for your valuable comments and looking forward to your reply.

---

> ### Author Response · Authors · 2022-12-02
> **A gentle reminder for Reviewer oVMw**
>
> Dear Reviewer oVMw,
>
> We appreciate your constructive comments for helping us improve our paper in many aspects. This is a gentle reminder since the final discussion stage ends soon.
>
> We would like to ask if Reviewer oVMw may have any further questions regarding our submission so that we can still respond. If you are satisfied with our response and changes, please consider updating the rating.
>
> Best wishes,
>
> Authors

---

### Official Review · Reviewer_69Fv · 2022-10-26

**Confidence:** 3
**Correctness:** 4
**Technical Novelty And Significance:** 4
**Empirical Novelty And Significance:** 4
**Recommendation:** 8

**Clarity, Quality, Novelty And Reproducibility:**

The proposed idea is original. The authors concisely describe the details of the method and the improvement from the existing methods.

**Strength And Weaknesses:**

Strengths
+ The map elements are represented as a set of points with a group of equivalent permutations which is suitable for expressing various shapes with edge directions.
+ The proposed hierarchical matching and loss functions look technically sound.
+ The performance is evaluated on public dataset and the proposed method achieves higher accuracy than the existing methods.
+ The transformation from 2D to BEV in map encoder could be substituted by other modules such as conventional IPM.

Weaknesses
- The experimental studies are carried out only on nuScenes and not on Argoverse2 dataset like the existing method.  Are there any reasons for this?
- More discussion on robustness towards real application is helpful.  For example, Are multi-camera input mandatory?  Can a front camera output HDMap in front of the vehicle?  How about the sensitivity of MapTR to camera position, intrinsic parameters, extrinsic parameters?  These robustness is important.

**Summary Of The Paper:**

The authors propose HD map construction algorithm from multi-view cameras.  They model each map element as a point set with a group of equivalent permutations.  The hierarchical matching both for instance-level and point-level is introduced and trained based on point2point loss and edge direction loss.  The experimental study shows that MapTR achieves higher extimation accuracy than the existing methods on public nuScenes dataset.  MapTR with light backbone can infer HD map at 25fps which is required for autonous driving.

**Summary Of The Review:**

The proposed method is novel and achieves definitely higher accuracy than the existing methods.  I think that this paper deserves acceptance to ICLR.

---

> ### Author Response · Authors · 2022-11-15
> **Thanks and Response to Reviewer 69Fv (2/2)**
>
>
> $\color{red}{Question~2:}$ More discussion on robustness towards real application is helpful.  For example, Are multi-camera input mandatory? Can a front camera output HDMap in front of the vehicle? How about the sensitivity of MapTR to camera position, intrinsic parameters, extrinsic parameters? These robustness is important.
>
> $\color{blue}{Response~2:}$ Thanks for your valuable suggestions. Multi-camera input is not mandatory for MapTR. MapTR is compatible with many sensor settings, e.g., 6-camera on nuScenes, 7-camera on Argoverse2, only LiDAR and camera & LiDAR. In Table R2, the experiments show that MapTR can achieve robust performance based on different sensor settings. And the experiments in Table R1 (6-camera on nuScenes and 7-camera on Argoverse2) show that MapTR is robust to camera number.
>
>
> | Method | Modality | Epochs| AP$_{ped}$|AP$_{divider}$|AP$_{boundary}$| mAP| FPS|
> | :---: | :---: |:---: |:---: |:---: |:---: |:---: |:---:|
> | MapTR-tiny   |Camera  | 24 |46.3|51.5|53.1|50.3|11.2
> | MapTR-tiny   |LiDAR  | 24 |48.5 |53.7|64.7|55.6|7.2
> | MapTR-tiny   |Camera&LiDAR  | 24 |55.9 |62.3|69.3|62.5|5.8|
>
> Table R2: Results of MapTR-Tiny based on different sensor settings.
>
> We agree with you that analysis about sensitivity and robustness is important. In real applications, the camera intrinsics are usually accurate and change little, but the camera extrinsics may be inaccurate due to the shift of camera position, calibration error, etc. To validate the robustness, we add random noise to the camera extrinsics and traverse the validation sets for evaluation of sensitivity and robustness. We respectively add translation and rotation deviation of different degrees. The noise is subject to normal distribution. As shown in Table R3 and R4, when the standard deviation of $\Delta_x, \Delta_y, \Delta_z$ is  $0.1\text{m}$ or the standard deviation of $\theta_x, \theta_y, \theta_z$ is  $0.02\text{rad}$, MapTR still keeps comparable performance.
>
> | $\sigma_{1}(\text{m})$ | 0 | 0.05| 0.1|0.5|1.0|
> | :---: | :---: |:---: |:---: |:---: |:---: |
> | mAP   |50.3  | 49.9|49.0|34.0|17.7|
>
> Table R3: Robustness of MapTR-tiny to the translation deviation of camera. $\sigma_1$ is the standard deviation of  $\Delta_x, \Delta_y, \Delta_z$.
>
>
> | $\sigma_{2}(\text{rad})$ | 0 | 0.005| 0.01|0.02|0.05|
> | :---: | :---: |:---: |:---: |:---: |:---: |
> | mAP   |  50.3 | 49.4|47.5|42.0|24.7|
>
> Table R4: Robustness of MapTR-tiny to the rotation deviation of camera. $\sigma_2$ is the standard deviation of  $\theta_x, \theta_y, \theta_z$.
>
> We hope the explanations and results above solve your concerns. Thanks for your valuable comments and looking forward to your reply.

---

> ### Author Response · Authors · 2022-11-15
> **Thanks and Response to Reviewer 69Fv (1/2)**
>
> Thanks for the valuable comments and suggestions which help us improve this work. Following are our detailed responses to your concerns.
>
> $\color{red}{Question~1:}$ The experimental studies are carried out only on nuScenes and not on Argoverse2 dataset like the existing method. Are there any reasons for this?
>
> $\color{blue}{Response~1:}$ Until the ICLR2023 submission deadline (Sep. 29th, 2023), the existing methods [1,2] only include experiments on nuScenes. Thus, we only report results on nuScenes following the setting of [1,2] for fair comparison. And thanks for your kind remind, we are currently working on applying MapTR on Argoverse2 benchmark to provide a baseline for the community. We attach our preliminary results in Table R1.
>
> We provide the processing details of Argoverse2 dataset here for comparison with nuScenes. nuScenes has 6 surround-view cameras and is annotated in 2Hz, with 28130 training samples. While Argoverse2 has 7 surround-view cameras and is annotated in 10Hz, with 109598 training samples. To keep the comparable training cost, we train MapTR on Argoverse2 for only 6 epochs. The 7 camera images of Argoverse2 have different resolutions ($1550\times2048$ for front view and $2048\times1550$ for others). To keep the consistent aspect ratio, we pad the 7 camera images into the same shape ($2048\times2048$), then resize the images with 0.3 ratio. The other settings are the same as nuScenes.
>
> As shown in Table R1, under the similar experimental settings, MapTR achieves impressive performance on both Argoverse2 and nuScenes.  Argoverse2 and nuScenes are collected from different regions all around the world. The consistent high performance proves the robustness and generalization ability of MapTR.
>
> We provide the code of MapTR in the supplementary materials for detail check. And we will release the code to the public soon.
>
>
> | Method | Dataset | Camera number | Training samples |Epochs|AP$_{ped}$|AP$_{divider}$|AP$_{boundary}$|  mAP|
> | :---: | :---: |:---: |:---: |:---: |:---: |:---: |:---:|:---:|
> | MapTR-tiny   |nuScenes | 6 | 28130 |24|46.3|51.5|53.1|50.3|
> | MapTR-tiny   |Argoverse2 | 7 | 109598 |6|57.9|56.9|59.2|58.0|
>
> Table R1: Results of MapTR-Tiny on Argoverse2 and nuScenes.
>
>
> [1]. Li, Qi et al. “HDMapNet: An Online HD Map Construction and Evaluation Framework.” 2022 International Conference on Robotics and Automation (ICRA) (2022).
>
> [2]. Liu, Yicheng et al. “VectorMapNet: End-to-end Vectorized HD Map Learning.” ArXiv:2206.08920 (2022).

---

### Decision · Program_Chairs · 2023-01-20

**Decision:**

Accept: notable-top-25%

**Justification For Why Not Higher Score:**

While the work is solid, it is unclear whether the work would be exciting for a broader set of researchers (e.g. those not doing vector map building or self-driving)


**Justification For Why Not Lower Score:**

All reviewers are positive on the work.  The AC agrees the work is solid and experiments show good performance.

**Metareview: Summary, Strengths And Weaknesses:**

Summary: The paper proposes a transformer-based method to construct high-definition (HD) vectorized maps from multi-view cameras.  These HD maps are essential for autonomous driving.  Each map element is represented as a point set (with different permutations to achieve a permutation-equivalent representation) with BEV features.  The map elements are matched in an hierarchical manner (point to point, instance to instance) with encoder-decoder model similar to DETR.  Experiments on nuScenes show the proposed method can both produce more accurate maps than prior work (HDMapNet and VectorMapNet) as well as run much faster.

Strengths:
- Reviewers find the work to be technically sound and sufficiently novel
- The method has demonstrated state-of-the-art results on nuScenes
- The proposed method is much faster than prior work and can produce HD maps in real time on RTX 3090

Weaknesses
- There is little discussion of the limitations of the work
- It's unclear whether the proposed method would work well for more complex / different types of maps than found in the self-driving domain

Most of the weaknesses noted by reviewers were addressed by the revision and author rebuttal

**Note From Pc:**

if the above contains the word "oral" or "spotlight" please see: "oral" presentation means -> notable-top-5% and "spotlight" means -> notable-top-25%. As stated in our emails, we are disassociating presentation type from AC recommendations